# Three-Dimensional Vulnerability Assessment of Peanut (*Arachis hypogaea*) Based on Comprehensive Drought Index and Vulnerability Surface: A Case Study of Shandong Province, China

Sicheng Wei [1,2,3], Yueting Yang [1,2,3], Kaiwei Li [1,2,3], Ying Guo [1,2,3] and Jiquan Zhang [1,2,3,*]

1   School of Environment, Northeast Normal University, Changchun 130024, China
2   State Environmental Protection Key Laboratory of Wetland Ecology and Vegetation Restoration, Northeast Normal University, Changchun 130117, China
3   Key Laboratory for Vegetation Ecology, Ministry of Education, Changchun 130117, China
*   Correspondence: zhangjq022@nenu.edu.cn; Tel.: +86-135-9608-6467

**Abstract:** Agricultural drought is a major problem facing China's agricultural production. In this study, the cash crop 'peanut' was used as an example to explore vulnerability. Through the atmosphere–plant–soil continuum system, a single index that could represent different types of droughts affecting peanuts was selected and weighted using the CRITIC weighting method to construct a multi-source data fusion drought index (MFDI). Then, Pearson correlation analysis between the comprehensive drought index and relative meteorological yield and the Mann–Kendall trend test for different growth periods of peanuts were used to verify MFDI and analyze the variation over time. A three-dimensional vulnerability assessment method of drought intensity–drought duration–yield reduction rate was established based on the run theory and trend surface analysis. The results show that the constructed multi-source data fusion drought index (MFDI) can more accurately characterize the actual drought conditions of peanuts in Shandong Province. The MFDI results showed that the drought severity in the coastal areas of the study area decreased with the growth and development of peanuts, while the drought became more severe in the western and northern parts during the late growth period of peanuts. The vulnerability surface of the drought intensity–drought duration–yield reduction rate showed that when the drought intensity was <0.8 and the duration was <3.5 months, the vulnerability of peanut crops was low, and then with the increase in drought intensity or duration, the vulnerability increased. The impact of drought duration cannot be ignored. In contrast to traditional vulnerability assessment methods, this study established a three-dimensional vulnerability surface, which provides a new approach for agricultural drought vulnerability assessment. The research results are helpful for a deeper understanding of the relationship between drought and crop vulnerability and provide scientific support for local governments in formulating disaster prevention and mitigation policies.

**Keywords:** vulnerability surface; CRITIC; multi-source data fusion drought index; peanut; agricultural drought vulnerability

## 1. Introduction

Currently, global warming is a serious problem facing humankind [1–3]. The impacts of climate change are unprecedented and irreversible [4]. An increase in temperature will lead to changes in precipitation and evapotranspiration, and then increase the frequency and intensity of drought, seriously threatening crop growth and affecting the development of the agricultural economy [5–7]. China is a country seriously affected by climate warming due to its variable terrain and complex geographical environment. Food security and agricultural economic development are significantly affected by drought disasters [8–10]. Consequently, it is a good strategy to identify the causes of drought vulnerability and evaluate it effectively to mitigate its impact on agriculture [11,12].

Peanut production is one of the pillar industries of agriculture, with significant economic benefits compared with other grains and oilseeds in the field. It is the best choice for agricultural structural adjustment and is crucial for agricultural economic development [13,14]. Shandong Province is a leading peanut-producing region in China in terms of planting area and production, with peanuts grown in all districts throughout the province [15]. However, in the context of climate change, the climate in the region tends to be warm and dry, with an evident increase in drought events. Drought leads to a reduction in photosynthetic leaf area, number of flowers per plant, effective flower number, and dry matter accumulation of peanuts, which suppresses peanut growth and significantly reduces economic output [16–18]. Due to the large fluctuations in peanut yield caused by drought, the stability of peanut production has been reduced, which has negatively impacted the economic development of Shandong Province. Therefore, it is imperative to strengthen the drought vulnerability assessment of peanuts and provide disaster prevention and mitigation policies to relevant local departments to maintain stable peanut production in Shandong Province.

In recent years, research into the drought effects on peanut crops has become increasingly detailed. Jiang et al. selected the precipitation anomaly percentage indicator to identify and explore drought effects on peanut crops and their distribution pattern. Based on this, they applied the natural disaster risk theory to assess the risk of peanut drought in Shandong Province [19]. Njouenwet et al. evaluated peanut drought in the Sudano–Sahelian region of Cameroon using a three-month standardized precipitation index in combination with peanut phenology [20]. Zhang et al. used the crop water deficit index considering the water demand of peanuts to identify drought in summer peanuts in Henan Province, clarified the temporal and spatial distribution characteristics of drought disasters in summer peanuts, and further evaluated the risk of peanut drought disasters [21]. However, the above indicators only focus on a single type of drought, without considering the growth characteristics of peanuts, which is not sufficient to characterize the real drought situation of peanuts in terms of comprehensiveness and accuracy. Therefore, it is necessary to construct a comprehensive drought index suitable for peanut drought based on a single index, from meteorological drought, soil drought, and the response of vegetation to drought, by integrating biological environment information and surface meteorological observation data [22] to achieve accurate identification of peanut droughts.

As a measure of the ability of a system to resist disasters, vulnerability assessment is a frequent focus of agrometeorological disaster risk assessment [23,24] and is an important link to determine the degree of damage to agricultural systems subject to disaster. Currently, there are three main methods of crop vulnerability assessment: (1) Vulnerability assessment based on an index system. When the vulnerability formation mechanism is not clear, indicators are selected based on sensitivity and adaptability to evaluate the crop vulnerability [25]. Wang et al. constructed an evaluation model of maize drought vulnerability by combining environmental sensitivity, exposure degree, crop sensitivity, and adaptability, and evaluated and analyzed maize drought vulnerability in the semi-arid areas of northwest China [26]. However, crop growth is a complex dynamic process and drought has different effects on crops at different growth stages. Therefore, this method cannot be used for vulnerability research at different growth stages. (2) Vulnerability assessment based on crop model. The crop model can simulate the growth and development processes of crops and quantify the effects on crop physiological indices caused by disasters more accurately. This method represents a new research direction for vulnerability assessment [27–29]. Pang et al. proposed a vulnerability curve calculation method based on the drought disaster intensity index combined with the CERES model and evaluated and zoned corn vulnerability in western Jilin [30]. However, owing to the different natural conditions in different regions, further research is needed to realize the transformation of vulnerability assessment from the field scale to the regional scale. (3) Vulnerability assessment based on vulnerability curves. Considering disaster intensity and crop loss rate, the most widely used vulnerability assessment method is to build a vulnerability curve [31–33]. Yang et al.

used the drought intensity and yield loss rate of millet to construct drought vulnerability curves of millet at different growth stages to dynamically evaluate the drought risk of millet in Liaoning Province [34]. However, the disaster loss rate is not a simple binary relationship with the disaster itself but is also related to many factors. Therefore, an increasing number of researchers are focusing on the characteristics of disasters and using the trend surface analysis method to build vulnerability surfaces to evaluate the vulnerability of crops. They aim to achieve regional vulnerability assessment, improve the accuracy of vulnerability assessment results, and narrow the gap with reality, which is a widely discussed topic in current vulnerability research [35].

In summary, the current indicators used to identify peanut droughts are relatively singular. Most of them use meteorological drought indicators as peanut drought identification indicators, which cannot accurately represent the actual drought stress of the peanut crop. In addition, research on the vulnerability to agrometeorological disasters is mostly focused on field food crops, using one- and two-dimensional perspectives. There is a lack of three-dimensional vulnerability assessment research on peanuts. Therefore, the objectives of this study are: (1) to combine peanut characteristics, select different types of drought indexes based on the atmosphere–crop–soil continuum theory, determine the weights of the these indexes using the CRITIC weighting method, construct a multi-source data fusion drought index, and judge the applicability of the index to peanuts; and (2) to determine the characteristics of peanut drought disaster by using run theory, construct a three-dimensional vulnerability surface based on the drought intensity–drought duration–yield reduction rate, and evaluate the drought vulnerability of peanut crops in Shandong Province. This study added remote sensing data to the calculation of peanut drought identification indicators and conducted a three-dimensional vulnerability assessment of regional peanut crops based on multi-data integration to better illustrate the relationship between drought and peanut crop vulnerability. The research results can help to understand the relationship between drought and peanut vulnerability more intuitively, provide a scientific basis for local governments to designate disaster prevention and mitigation policies, and provide new ideas for peanut vulnerability assessment.

## 2. Materials and Methods

*Study Area and Data Sources*

As a coastal province in eastern China, Shandong extends between $34°22.9'$–$38°24.01'$N and $114°47.5'$–$122°42.3'$E. It covers an area of $1.56 \times 10^5$ km$^2$. The area of agricultural land in Shandong Province is $1.16 \times 10^5$ km$^2$, accounting for 73.6% of the total land area. The study area has a warm temperate monsoon climate, with an annual average temperature of 11–14 °C, and annual precipitation of 550–950 mm, which decreases from southeast to northwest. An overview of the research area is shown in Figure 1. The planting area of peanuts is approximately $7 \times 10^3$ km$^2$. The crop is usually sown in May and harvested in September [36].

The data used in this research include meteorological, remote sensing, soil, agricultural, and disaster data. Specific information is shown in Table 1. The daily precipitation, temperature, sunshine duration, wind speed, and average relative humidity of 22 stations from 1991–2020 were used to calculate the standardized precipitation evapotranspiration index (SPEI). Ten-day NDVI data were used to calculate vegetation condition index (VCI). The monthly soil water content data were used to calculate the soil moisture status index (SMCI). All the above calculations were carried out using MATLAB 2018. Meteorological data were used to calculate the SPEI, and the inverse distance weight (IDW) method in ArcGIS was used for interpolation to obtain a spatial distribution map of the SPEI with an average of 30 years in the study area. Remote sensing data were used to calculate the VCI and SMCI, and the raster calculator and mask extraction method in ArcGIS were used to obtain the spatial distribution map of the VCI and SMCI with an average of 30 years in the study area.

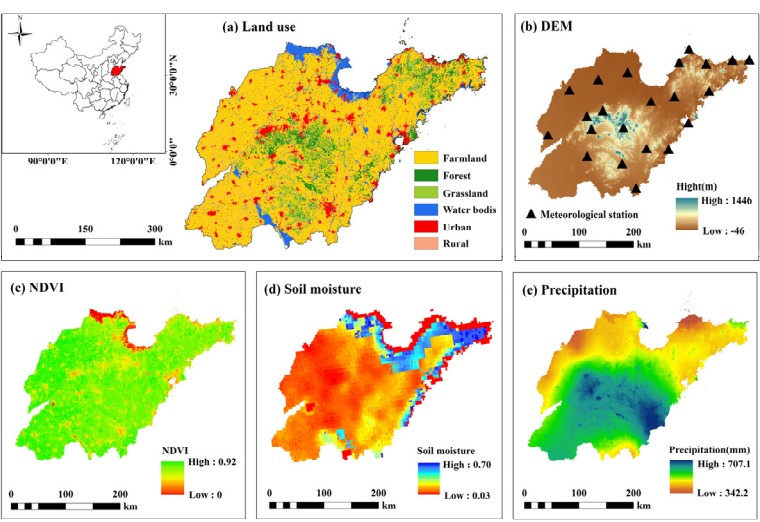

**Figure 1.** Overview of the research area.

**Table 1.** Data type and sources used in this study.

| Data Type | Data Contents | Resolution | Data Sources (1991–2020) |
|---|---|---|---|
| Daily meteorological data | Daily precipitation, Temperature, sunshine duration, wind speed and average relative humidity | 22 meteorological stations in Shandong Province | Meteorological Data Center of China Meteorological Administration (http://data.cma.cn/site/index.html, accessed on 10 May 2021) |
| Remote sensing data | Ten-day NDVI | 500 m | Computer Network Information Center International Scientific Data Mirror Website (http://www.gscloud.cn, accessed on 10 May 2021) |
| Soil data | Monthly soil moisture | 4 km × 4 km | Terra climate data sets (http://www.climatologylab.org, accessed on 10 May 2021) |
| Agricultural data | Peanut yield and sown area | Districts in Shandong Province | Institute of Meteorology, Department of Planting Management, Ministry of Agriculture, China (http://www.moa.gov.cn/, accessed on 10 May 2021) |
| | Data of peanut development period | | |
| Other data | Historical disaster data | Districts in Shandong Province | Disaster Occurrence and China's agricultural statistics |
| | Basic data of research area | | China Statistical Yearbook |

## 3. Methodology

### 3.1. Construction of Comprehensive Drought Index Based on the Atmosphere–Plant–Soil Continuum

Peanut, as a typical temperate and high-water-demand crop, faces different types of droughts in the process of growth and development, such as meteorological and soil droughts [19]. Crops respond differently to different types of droughts; therefore, it is essential to select indexes that can better characterize various droughts. From the perspective of the atmosphere–plant–soil continuum, this paper selects SPEI, VCI, and SMCI to construct a comprehensive drought index (MFDI) in order to describe the drought effects on peanut crops. According to previous research [37], the growth period can be divided into three stages: the early growth period (May), middle growth period (June), and late growth period (July–September).

### 3.1.1. Standardized Precipitation Evapotranspiration Index

SPEI is a drought index proposed by Vicente-Serrano et al. that considers precipitation and evapotranspiration [38]. It can reflect the drought situation due to differences in

precipitation and evapotranspiration in different areas during different periods. The calculation of SPEI mainly uses meteorological data, such as monthly precipitation and average temperature, and is obtained by calculating the difference between precipitation and evapotranspiration and normalizing the difference [39]. The specific calculation steps are as follows.

(1) Calculate the difference between precipitation and potential evapotranspiration:

$$D_i = P_i - PET_i \tag{1}$$

where $D_i$ is the difference between precipitation and evapotranspiration in the *ith* growth period (mm), $P_i$ is the precipitation in the *ith* growth period (mm), and $PET_i$ is the potential evapotranspiration (mm) in the *ith* growth period, which is calculated by the Penman–Monteith method recommended by the FAO (Food and Agriculture Organization of the United Nations, Rome, Italy) [40].

(2) The log-logistic probability distribution function is used to normalize the D data series, and the SPEI value corresponding to each numerical value is calculated:

$$SPEI = w - \frac{c_0 + c_1 w + c_1 w^2}{1 + d_1 w + d_2 w^2 + d_3 w^3} \tag{2}$$

$$w = \sqrt{-2 \ln P} \tag{3}$$

where $p$ is the cumulative probability of exceeding the undetermined D value, and when $p > 0.5$, the sign of the SPEI value is reversed; $c_0 = 2.515517$, $c_1 = 0.802853$, $c_2 = 0.010328$, $d_1 = 1.432788$, $d_2 = 0.189269$, and $d_3 = 0.001308$.

### 3.1.2. Vegetation Condition Index

VCI reflects the impact of drought disasters on vegetation at different growth stages of peanuts [41]. The formula used is as follows:

$$VCI_i = \frac{NDVI_i - NDVI_{min}}{NDVI_{max} - NDVI_{min}} \tag{4}$$

where *VCI* represents the vegetation growth status and $NDVI_i$ is the index value of each pixel. $NDVI_{max}$ and $NDVI_{min}$ are the maximum and minimum NDVI of each pixel in each growth period over the last 30 years, respectively.

### 3.1.3. Soil Moisture Condition Index

In order to quantify the profit and loss of soil moisture in the peanut crop root zone, the following formula is suggested for SMCI [42]:

$$SMCI_i = \frac{SM_i - SM_{min}}{SM_{max} - SM_{min}} \tag{5}$$

where *SMCI* stands for the standardized soil moisture status index, and $SM_i$ is the pixel value of soil moisture in the root zone during the same month of the last 30 years. $SM_{max}$ and $SM_{min}$ represent the maximum and minimum values of soil moisture in each pixel of each growth stage over the last 30 years, respectively.

### 3.2. CRITIC Weighting Method

In comparison with the entropy and standard deviation methods, the CRITIC weighting method is a more objective weighting method. According to this method, the objective weights of the indicators are measured by the contrast strength and conflict. It considers the variability of indicators and the correlation between indicators, and makes full use of the objective attributes of the data [43]. The contrast intensity is expressed in the form of the standard deviation. The greater the standard deviation, the greater the fluctuation and the higher the weight. The conflict between the indicators is expressed using the correlation

coefficient. A strong positive correlation between the two indicators implies a lower weight if the conflict is smaller [44].

The specific method is as follows [45]:

Assuming n = evaluation samples and m = evaluation indicators, the original indicator data matrix is formed as:

$$X = \begin{pmatrix} x_{11} & \cdots & x_{1m} \\ \vdots & \ddots & \vdots \\ x_{n1} & \cdots & x_{nm} \end{pmatrix} \tag{6}$$

where $X_{ij}$ represents the value of the *ith* sample and the *jth* evaluation index.

(1) Dimensionless treatment

To eliminate the influence of different dimensions on the evaluation results, it is necessary to perform a dimensionless treatment on each index. In this study, a smaller index value indicates drier conditions; therefore, the dimensionless treatment is carried out using the following formula:

$$X' = \frac{x_{max} - x_j}{x_{max} - x_{min}} \tag{7}$$

(2) Index variability

The index variability is expressed as a form of the standard deviation:

$$\begin{cases} \overline{x_j} = \frac{1}{n} \sum_{i=1}^{n} x_{ij} \\ S_j = \sqrt{\frac{\sum_{i=1}^{n} (x_{ij} - \overline{x_j})^2}{n-1}} \end{cases} \tag{8}$$

where, $S_j$ represents the standard deviation of the *jth* index.

The standard deviation of the CRITIC weight method measures the fluctuation of the internal values of each index. By increasing the standard deviation, the numerical difference in the index increases, revealing greater information and increasing the level of evaluation intensity. Therefore, more weights were assigned to the index.

(3) Index conflict

The index conflict is expressed by the correlation coefficient:

$$R_j = \sum_{i=1}^{m} (1 - r_{ij}) \tag{9}$$

where, $R_{ij}$ represents the correlation coefficient between evaluation indices *i* and *j*.

The correlation coefficient measures the relationship between two indicators. When the two indicators are highly correlated, there is little conflict between them. In other words, the weights assigned to the indicators should be reduced because they reflect the same information.

(4) Quantity of information

$$C_j = S_j \sum_{i=1}^{m} (1 - r_{ij}) = S_j \times R_j \tag{10}$$

The larger the value of $C_j$, the greater the role of the *jth* evaluation index in the entire evaluation index system, and the more weight that should be assigned to it.

(5) Objective weight

The objective weight $W_j$ of the *jth* index is:

$$W_j = \frac{C_j}{\sum_{j=1}^{m} C_j} \tag{11}$$

### 3.3. Construction of Multi-Source Data Fusion Drought Index

Because the spatial resolution was different among the indicators, resampling was performed first to change the spatial resolution of each dataset to 4 km × 4 km. MFDI was constructed using the CRITIC weighting method, and the calculation method is as follows:

$$MFDI_i = w_1 \times SPEI_i + w_2 \times VCI_i + w_3 \times SMCI_i \tag{12}$$

where $w_1$, $w_2$, and $w_3$ are the weight values of the three indices.

### 3.4. Relative Meteorological Yield Reduction Rate

Generally, three main categories of factors affect crop yield formation: meteorological conditions, agronomic and technological measures, and stochastic factors. Technical measures of agricultural production are a measure of the level of development of social production over time. Short-term technology trends are referred to as trend outputs, and meteorological production reflects the components of short-term yields affected by meteorological factors. A small proportion of calculations is affected by stochastic factors, which are often ignored [46,47] so that:

$$Y = Y_t + Y_w \tag{13}$$

where, $Y$ is the actual yield (single production) of the crop, $Y_t$ is the trend yield, and $Y_w$ is the meteorological yield.

In this study, the trend yield was simulated using the straight-line sliding average method. It is a method of modeling yield which considers the change in yield within a given stage as a linear function, depicting a straight line. As the stage continually slides, the straight line constantly changes positions, and the backward slip represents continuous changes in the yield evolution trend. A regression model was obtained at each stage, and its trend yield value was considered as the mean of the linear sliding regression simulations at each time point [38]. The linear trend equation at a given stage is:

$$Y_i(t) = a_i + b_i t \tag{14}$$

where $i = n - k + 1$ and is the number of equations, $k$ is the sliding step, n is the number of sample sequences, and t is the time serial number. $Y_{i(t)}$ is the function value of each equation at point $t$ with $q$ function values at point $t$. The number of q is related to n and k. To calculate the average value of each function value at each point:

$$\overline{Y_i(t)} = \frac{1}{q} \sum_{j=1}^{q} Y_i(t) \tag{15}$$

The historical evolution trend of production can be observed by connecting the $\overline{Y_i(t)}$ value of each point. These characteristics differ depending on the value of k. Trend production can only eliminate the effects of short-term fluctuations at a large value of k. In this study, we take k to be five, based on the length of the production series after comparison.

After the yield trend was obtained, the meteorological yield was calculated using Equation (16), and the relative meteorological production is:

$$Y_r = \frac{Y_w}{Y_t} \tag{16}$$

Relative meteorological yields indicate that the variability in yield fluctuations that deviate from the trend (i.e., the amplitude of yield fluctuations) is not affected by time and space. On the other hand, a negative value indicates unfavorable meteorological conditions for crop production, that is, crop yield reductions.

### 3.5. Mann–Kendall Trend Test

The Mann–Kendall trend test is a method of diagnosing and predicting climate change by identifying when a climate mutation occurs in a series of climate data [48]. The World Meteorological Organization (WMO) recommends this as a nonparametric statistical test method. Additionally, the test can be used to test the significance of a long time series. Unlike conventional methods, this method requires no sample distribution and does not consider outliers. Meanwhile, sequence and type variables can be quantified, detected, and calculated easily [49,50]. For time series x with n samples, we constructed an order column:

$$S_k = \sum_{i=1}^{k} r_i, \ k = 2, 3, \ldots, n \tag{17}$$

The value of $r_i$ in the formula is as follows:

$$r_i = \begin{cases} +1 & x_i > x_j \\ +0 & x_i \le x_j \end{cases} \quad j = 1, 2, \ldots, i \tag{18}$$

The order column $S_k$ is the cumulative number of values at moment $i$ that is greater than the value at moment $j$. When $x_1, x_2, \ldots, x_n$ are independent of each other and have the same continuous distribution, the mean value $E(S_k)$ and variance $var(S_k)$ of $S_k$ are calculated:

$$\begin{cases} E(S_k) = \dfrac{k(k-1)}{4} \\ var(S_k) = \dfrac{k(k-1)(2k+5)}{72} \end{cases} \quad k = 2, 3, \ldots, n \tag{19}$$

Statistics are defined under the assumption of random independence of the time series:

$$UF_k = \frac{[S_k - E(S_k)]}{\sqrt{var(S_k)}} \quad k = 1, 2, \ldots, n \tag{20}$$

where $UF_1 = 0$, and $UF_k$ is the standard normal distribution, which is the statistical sequence calculated according to the time series sequence $x_1, x_2, \ldots x_n$. Then, according to the reverse order of the time series $x_n, x_{n-1}, \ldots x_1$, the above process is repeated, and $UB_k = -UF_k \times (k = n, n - 1, \ldots, 1)$, $UB_1 = 0$. Given the significance level $\alpha$, if $\alpha = 0.05$, the critical value $\mu_{0.05} = \pm 1.96$.

If the $UF_k$ line changed to the critical line in the test curve, and the trend and mutation of the change curve were not obvious. The value of $UF_k$ was greater than zero, which indicated that the sequence exhibited an upward trend, whereas it exhibited a downward trend. When it exceeded the critical line, it indicated a significant upward or downward trend. If the two curves $UF_k$ and $UB_k$ intersected at the critical line, the moment corresponding to the intersection was the time at which mutation began.

### 3.6. Run Theory

"Run-length" refers to a series of the same variable satisfying certain conditions in a sequence with finite values. The number of times the same variable appears is called the run length. At present, run theory is widely used in research on meteorological droughts. Figure 2 shows a conceptual diagram of event recognition based on the run theory.

When using run theory to identify drought and flood events, first, an interception level is given according to the grading standard of drought indicators, and the discrete series changing with time is intercepted. When the disaster index is lower than a certain threshold and the duration exceeds a certain length, a disaster event is considered to have occurred [51]. When the random variable is continuously greater than the interception level one or more times, a positive run occurs, and vice versa, a negative run occurs. In the process of identifying drought runs, the length of the negative run is called drought duration, and drought intensity is the area covered by the drought duration and interception

level [52]. According to the grade of drought indicators, the index in the time series is separated into two characteristic variables, the duration and intensity of drought events, using run theory.

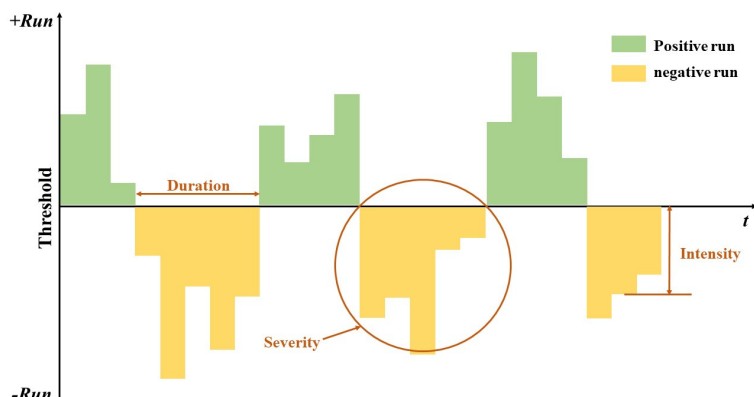

**Figure 2.** Schematic diagram of run theory.

*3.7. Vulnerability Surface Theory*

Vulnerability is a dynamic quantity, and a key physiological indicator of crop changes through changes in disaster intensity and duration. To evaluate vulnerability effectively, we need not only regular evaluations but also multifaceted approaches using constant processes and real-time monitoring. Research objectives in most vulnerability assessments are centered on establishing vulnerability curves, and the establishment of vulnerability curves is generally limited to two dimensions: damage intensity and loss. However, in reality, vulnerability is affected by different scales, so it is difficult for the two-dimensional vulnerability curve to express the continuous spatial variability of vulnerability. Based on the above aspects, building a three-dimensional vulnerability surface, considering the disaster duration, can express the continuous spatial variability of vulnerability and improve the spatial accuracy of assessment.

A vulnerability surface which simulates the changes in disaster intensity, disaster duration, and loss rate was generated by trend surface analysis. The construction of the vulnerability surface was based on a binary nonlinear regression analysis, and the basic equation is as follows [34,35]:

$$f(x,y) = \widetilde{f}(x,y) + \varepsilon \tag{21}$$

$$\widetilde{f}(x,y) = a + bx + cy + dx^2 + exy + fy^2 \tag{22}$$

where $a$, $b$, $c$, $d$, $e$, and $f$ are the coefficients estimated using the least-squares method. The fitting function is then used as the analytical expression for the vulnerability surface model:

$$Z = V(x,y) = \widetilde{f}(x,y) \tag{23}$$

where $Z$ is the yield reduction rate and $X$ and $Y$ represent the disaster intensity and disaster duration, respectively.

The vulnerability surface was drawn by a curve-fitting module in MATLAB, where the $X$, $Y$, and $Z$ axes represent drought intensity, drought duration, and peanut yield reduction rate, respectively. The average drought index of farmland in each city was chosen as the drought intensity, and the vulnerability surface was constructed according to the peanut yield reduction rate and drought duration in the city.

## 4. Results and Discussion

*4.1. Single Drought Index Analysis*

The 30-year average SPEI, VCI, and SMCI of the peanuts during each growth period are shown in Figure 3. Over the entire development stage of peanuts, the SPEI in the

study area is distributed in the range of −0.21–0.22, which indicates that the meteorological conditions in the study area were generally favorable for the growth and development of crops. Within this range, the spatial distributions of SPEI in the early and middle growth periods of peanut were found to follow a similar pattern. Dezhou, Weifang, and Linyi were wet, whereas some areas in Yantai and Jinan were slightly dry. The distribution of the VCI at each growth stage was different. In the early growth period, it was humid in the west but turned humid in the east during the middle growth stage. At a later stage of growth, the entire study area showed a relatively humid trend. For SMCI, Dezhou, Yantai, and Qingdao were relatively wetter during the early growth stage of peanuts, and the northwest of the entire research area was in a wet state during the middle growth stage. The difference is that, in the late growth period of peanuts, the humid areas seemed to be concentrated in the southeastern part of the study area.

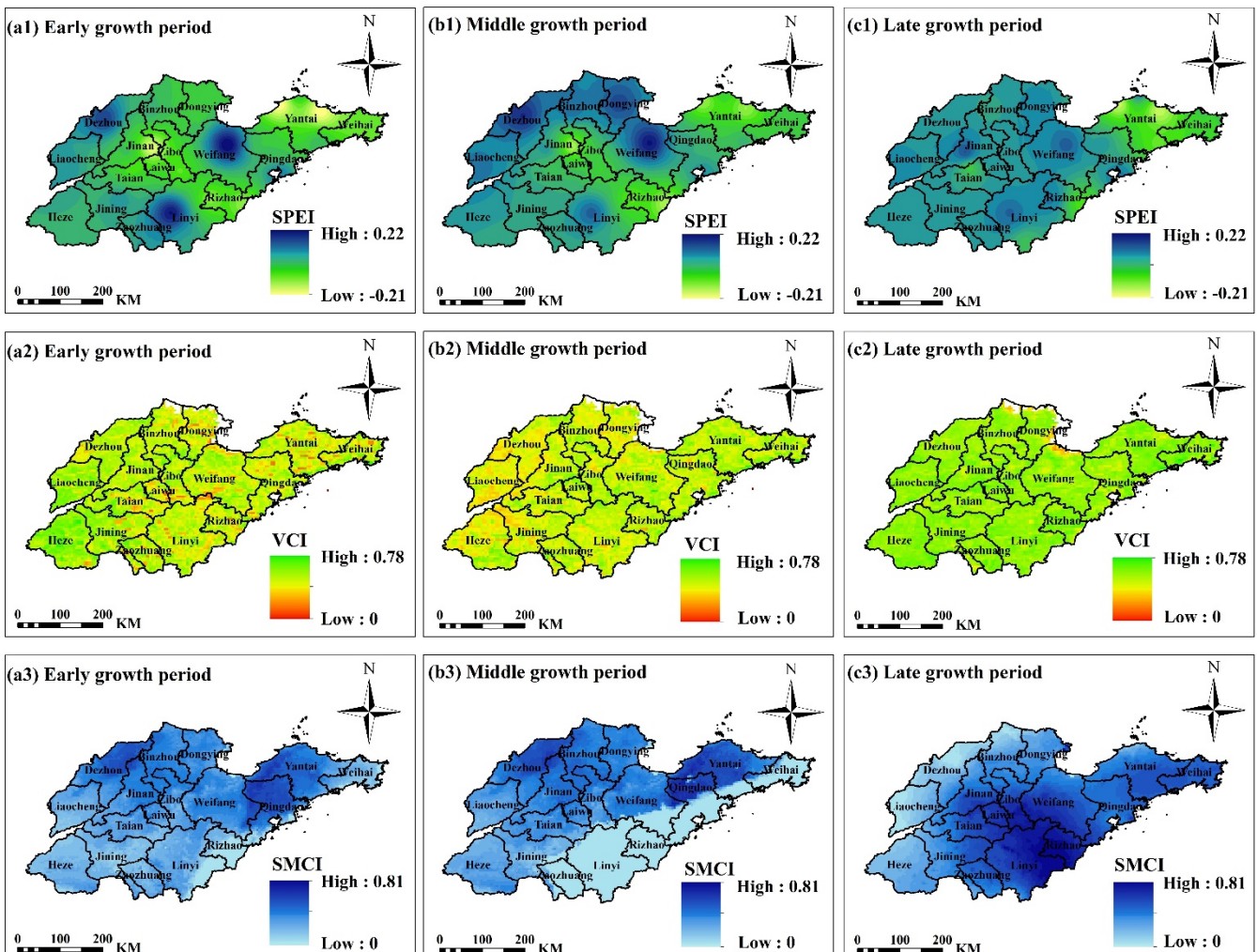

**Figure 3.** Spatial distribution of single drought index ((**a**), (**b**) and (**c**) represent early, middle, and late growth periods, respectively; (**1**), (**2**) and (**3**) represent SPEI, VCI, and SMCI, respectively).

Many studies on crop drought in earlier years used the SPEI as an evaluation standard and assumed that meteorological drought would lead to crop drought [53,54]. However, meteorological drought does not reflect the drought situation of the crops themselves; rather, it reflects the meteorological situation of this region, and the crops themselves may be affected by more factors. It is a widely accepted method to use VCI as an index to evaluate drought when assessing the condition of the crops themselves. Under water stress, crops change the opening degree of stomata to cope with water gain and loss. At this time, NDVI can capture such situations. Vegetation cover also affects crop conditions. Vegetation

cover may affect the light area, and the light area changes the soil temperature, which affects the evapotranspiration of crops. [55,56]. In this regard, the ability of NDVI to capture vegetation cover is also an important reason why it is often selected. Soil is a direct channel for crops to grow and obtain nutrients through the roots. The soil provides direct access to nutrients and aids in the growth of crops; therefore, the influence of soil moisture on crops is self-evident [57,58]. When peanuts, a crop whose fruit grows in the ground, begin to grow, the soil is constantly in contact with the fruit until harvest. Therefore, it is reasonable to choose the SMCI as an index to judge the degree of drought.

Through spatial and temporal analysis of the single drought index, the spatial drought distributions of the three single drought indices seem to be different. This shows that, regardless of the meteorological drought, crop drought, or soil drought alone, when evaluating the long-term drought situation in a region, the results obtained may not accurately reflect the real state of crops.

### 4.2. Establishment and Analysis of a Drought Index Based on Multi-Source Data Fusion

The comprehensive drought index weights obtained using the CRITIC evaluation method are listed in Table 2.

**Table 2.** Weight of each single index in different growth stages of peanut.

| Comprehensive Drought Index | Peanut Growth Period | | |
| --- | --- | --- | --- |
| | Early Growth Period | Middle Growth Period | Late Growth Period |
| Meteorological drought (SPEI) | 0.31 | 0.33 | 0.17 |
| Vegetation drought (VCI) | 0.34 | 0.32 | 0.37 |
| Soil drought (SMCI) | 0.35 | 0.35 | 0.46 |

### 4.2.1. Drought Index Verification

The Pearson correlation coefficients between MFDI and meteorological yield during the peanut growing season in major peanut-producing areas of the study area during 1991–2020 are shown in Figure 4. In the time distribution, the correlation between MDFI and meteorological yield seems to have significantly improved with the growth and development of peanuts. Except for a few districts, MFDI in the middle and late growth stages of peanuts was positively correlated with meteorological yield in most districts. Figure 4 shows that the correlation degree between MFDI and the meteorological yield of peanuts in Jinan, Taian, and Weifang at the late growth stage is high, with values passing the significance test ($p \leq 0.01$). In summary, the MFDI index has a good ability to characterize the relative meteorological yield of peanuts; that is, MFDI index can characterize the drought characteristics of peanuts in this area.

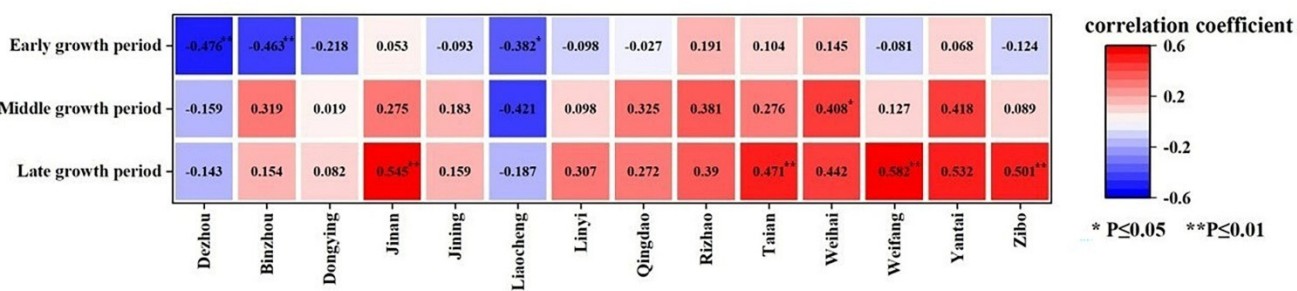

**Figure 4.** Pearson correlation analysis between MFDI and peanut yield reduction rate in different districts.

### 4.2.2. MFDI Time Series Analysis

Figure 5 shows the trend of MFDI in the study area over the past 30 years, along with the results of the M–K test. Early in the peanut growth stage, the UF and UB curves crossed in 2002, which was considered a mutation year, but the mutation was not significant.

As shown in the UF curve, before 2005, the index was less than zero, indicating that the regional crops were arid. Subsequently, the index was greater than 0, indicating an upward trend and wetting. According to historical records, 2005 was the third highest rainfall year in Shandong since 1999 owing to heavy rain and was a typical flood year [59]. During the middle period of peanut growth, from 2008–2016, the index increased. In 2008, the summer temperature was low and there was high precipitation; in 2016, the average annual temperature was the highest since 1951 [60,61]. As peanut crops grew, the UF curve was usually higher than 0, indicating that they were wet in many years. In general, through linear regression, peanuts showed drought effects at different growth stages, with rates of $-0.012$ units/10a, $-0.072$ units/10a, and $-0.014$ units/10a, respectively.

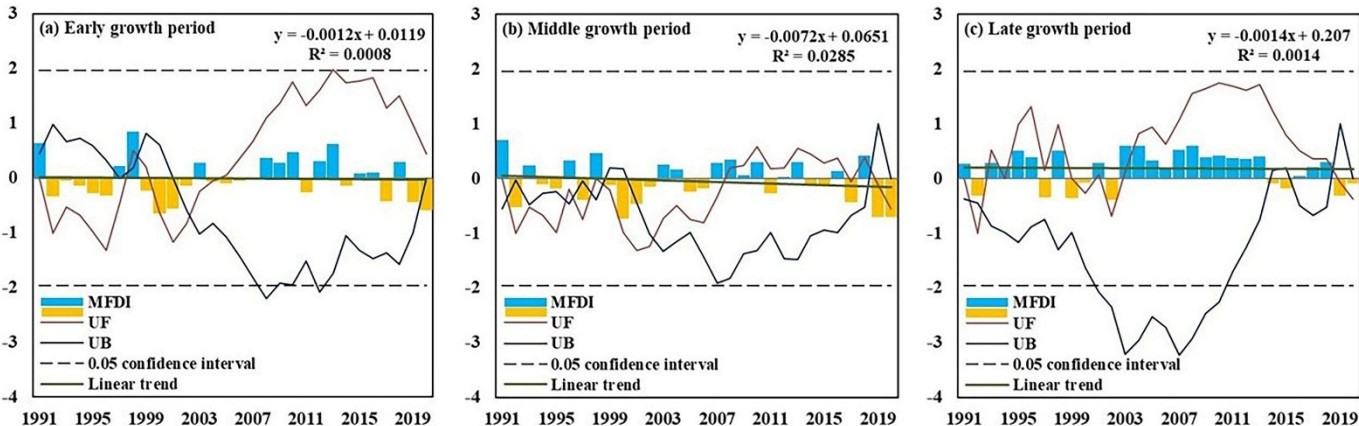

**Figure 5.** M–K test of MFDI in different growth stages ((**a**), (**b**) and (**c**) represent early, middle, and late growth periods, respectively).

### 4.2.3. MFDI Spatial Distribution

The MFDI of peanuts at different growth stages in the study area is shown in Figure 6. During the early stages of peanut growth, drought was severe in the northern part of the study area (Weihai and Rizhao). During the middle growth period of peanuts, the entire coastal area became arid, except for scattered places in the northern part of the study area. During the later stage of peanut growth, the drought trend appeared to reverse, with a wide range of drought in the northwest of the study area, but a humid situation in the area centered on Rizhao.

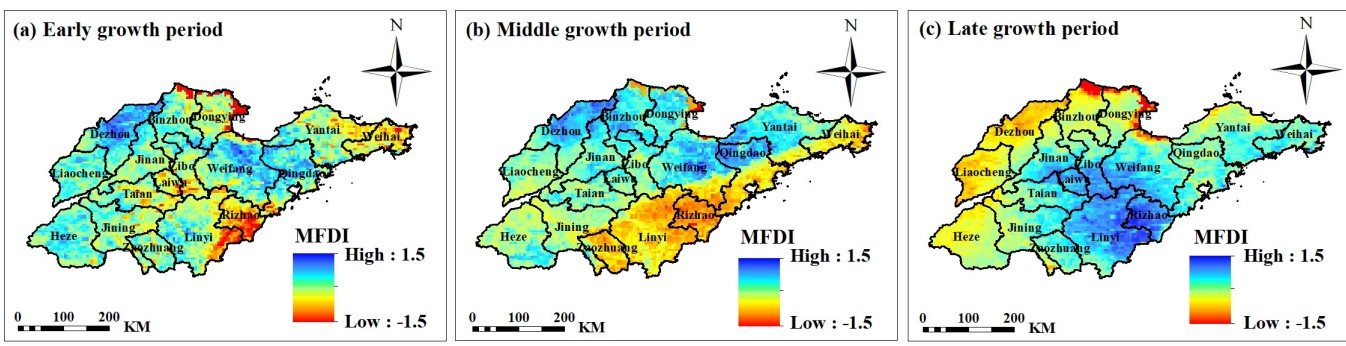

**Figure 6.** MFDI spatial distribution ((**a**), (**b**) and (**c**) represent early, middle, and late growth periods, respectively).

In summary, the spatial distribution of MFDI during the entire growth stage of peanuts was consistent with the results of Jiang et al. [19]. Drought effects on peanuts occurred mainly in the northwest of Shandong Province and the Jiaodong Peninsula. In addition, the M–K test for MFDI could also identify the mutation of drought in Shandong Province, indicating that the MFDI index could accurately characterize the effects of drought on

peanut crops in Shandong Province and could provide guidance for future studies on peanut drought.

As previously discussed, a single crop drought index may not adequately reflect the actual drought situation across the entire growth stage of crops. Some researchers have also attempted to evaluate the drought situation more precisely by developing a comprehensive drought index [23,62], but there are generally two problems: the selection of indicators and methods of empowerment. At present, the concept of the atmosphere–plant–soil continuum is widely used in agricultural drought research; therefore, three different indicators, namely, atmosphere-SPEI, plant-VCI and soil-SMCI, were selected to characterize the drought situation of peanut crops, which is more rigorous from the perspective of index selection. For a comprehensive drought index, the weighting of each index is always the most crucial factor. Initially, some studies adopted a subjective analytic hierarchy process. In this process, different experts' views on the same situation may differ significantly. In addition, the degree of expertise and number of experts also have a significant influence on the weighting. Later, some researchers began to examine objective evaluation methods, such as the entropy weight method or a combination of several weighting methods [24,25]. However, owing to objective problems in statistics, the results may be quite different from the real situation. Based on a large number of comparative analyses, this study chose the CRITIC weighting method to weigh the different drought indicators. The CRITIC weighting method is better than the entropy weighting method because it comprehensively measures the contrast and conflict intensities of each indicator, makes full use of the objectivity between indicators, and minimizes deviation from the real situation. This method is rarely used in agricultural index weighting but should be further investigated.

### 4.3. Analysis of Drought Characteristics Based on Run Theory

Using run theory, the duration, intensity, and frequency of drought in the study area over the past 30 years were obtained (Figure 7). The drought characteristics in different areas were clearly seen. As shown in Figure 7a, the areas with long drought durations were concentrated in the west of the study area, among which Binzhou, Liaocheng, and Jining had the longest drought durations. Figure 7b shows that the overall drought intensity in the eastern part of the study area was low, and Yantai and Qingdao had the lowest drought intensity. Areas with high drought intensity were Jining, Binzhou, and Weifang. Figure 7c shows the distribution of the drought frequency. The areas with the highest drought frequency were Weifang and Weihai, and the drought frequency in the western part of the study area was also high, whereas the drought frequency in Linyi, Jinan, and Binzhou was low.

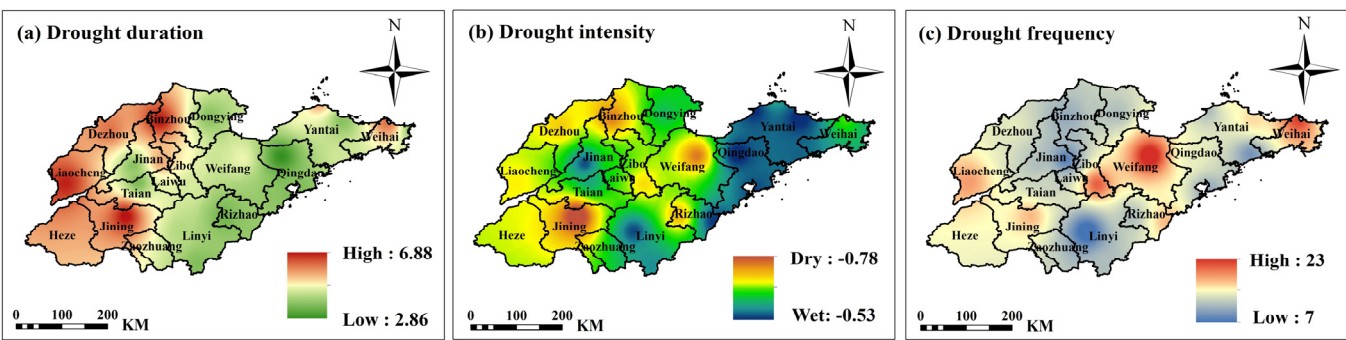

**Figure 7.** Spatial distribution of drought characteristics in the study area ((**a**), (**b**) and (**c**) represent drought duration, intensity, and frequency, respectively).

### 4.4. Vulnerability Surface Analysis

Based on the drought duration, drought intensity, and relative meteorological yield reduction rate of peanuts in different areas over 30 years, a vulnerability surface map of peanuts during the growth and development stages in the study area was constructed.

According to the vulnerability surface (Figure 8), the peanut vulnerability seems low when the drought intensity is < 0.8 and the duration is < 3.5 months. With an increase in drought intensity or duration, the peanut crop vulnerability increased. At this time, when the drought intensity remained the same, the peanut yield loss rate gradually increased with increasing drought duration. Similarly, when the duration of drought remained the same, the increase in drought intensity significantly increased peanut crop vulnerability. These results indicate that the influence of drought duration cannot be ignored. Based on the vulnerability surface, the specific vulnerability can be intuitively visualized in relation to the duration and intensity of droughts.

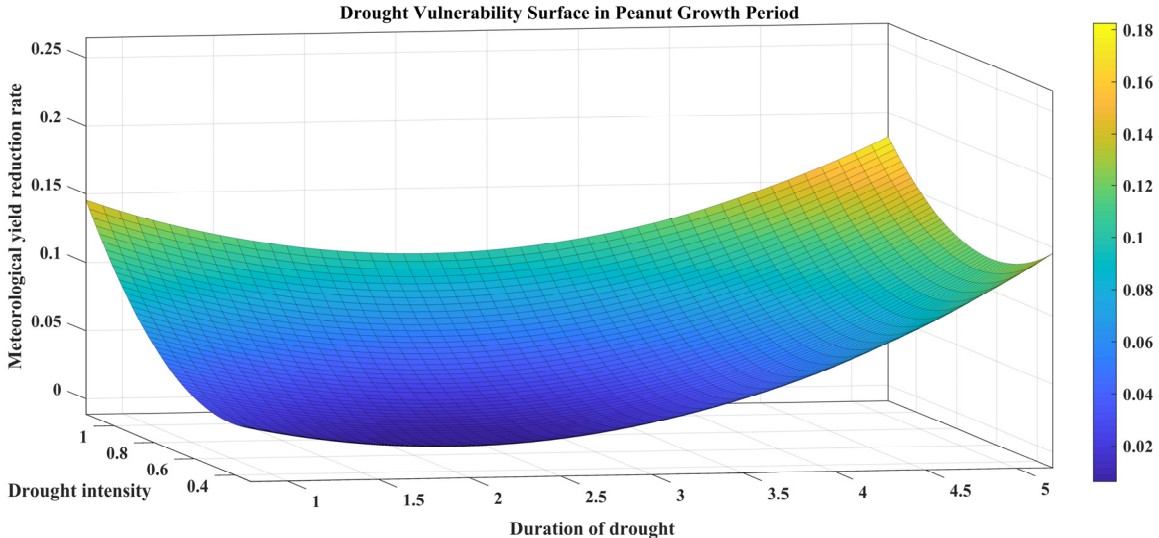

**Figure 8.** Drought vulnerability surface during peanut growth stage.

Researchers have found a correlation between the crop yield loss rate and drought intensity through changes in water status [31]. These curves can enable managers and researchers to obtain drought information and provide a reference for assessing crop drought risks. However, it is pertinent to note that crop yield loss is not only influenced by the intensity of the drought but also by the duration of the drought. As a result, establishing a relationship surface between drought intensity, drought duration, and crop yield loss rate provides managers with more comprehensive information to adjust irrigation policies accordingly.

### 5. Conclusions

This study presented a three-dimensional vulnerability evaluation method based on the disaster intensity–disaster duration–yield reduction rate for the first time in the drought risk assessment of peanut crops, which reflected the spatial variability of vulnerability and greatly improved the accuracy of evaluation. By rationally selecting the atmosphere–plant–soil index and applying the CIRTIC weighting method to create a comprehensive drought index, the accuracy of the vulnerability evaluation was further improved, which provides a novel concept of agricultural vulnerability evaluation. In future studies, we may be able to make more accurate predictions by combining crop models and machine learning. From the research results, we can observe the following: (1) In the study area, soil drought was the most important factor affecting peanut growth and development. (2) The constructed multi-source data fusion drought index was superior to each single index (SPEI, VCI, and SMCI) in identifying peanut drought and could more accurately represent the actual situation of peanut drought in Shandong Province. (3) The variation over time in MFDI showed that drought severity was aggravated with the passage of time in different growth stages of peanut crops. (4) The MFDI identification results revealed that the degree of peanut drought in the coastal areas of the study area reduces with the

development of peanuts, whereas the western and northern parts of Shandong Province tend to experience drought during the late growth period of peanuts. (5) The vulnerability surface based on the drought intensity–drought duration–yield reduction rate considered the compound relationship between drought intensity and duration, which overcomes the limitations of traditional vulnerability evaluation methods (i.e., one-dimensional index systems and two-dimensional vulnerability curve evaluation methods). Considering the vulnerability surface, it can be seen that when the drought intensity was < 0.8 and the duration was < 3.5 months, the vulnerability of peanuts was low, and then with the increase in drought intensity or duration, the vulnerability of peanuts increased. At the same time, drought intensity and drought duration were considered as the influencing factors of peanut crop vulnerability. Trend surface analysis was used to construct a peanut drought vulnerability surface, which is a novel idea and method for vulnerability assessment and has great potential for application in other research fields or other crops. However, it is worth noting that when this method is applied to other research fields or crops, it is necessary to re-select the indicators, calculate the weights according to the actual situation, and verify them to evaluate the vulnerability of crops more accurately and comprehensively.

**Author Contributions:** Conceptualization, S.W. and K.L.; Data curation, S.W. and Y.Y.; Formal analysis, Y.Y.; Investigation, S.W.; Methodology, S.W. and Y.Y.; Resources, K.L. and Y.G.; Software, Y.Y. and Y.G.; Supervision, J.Z.; Writing—original draft, S.W.; Writing—review and editing, J.Z.; Funding acquisition, J.Z. All authors have read and agreed to the published version of the manuscript.

**Funding:** This study was supported by the National Key R&D Program of China (2019YFD1002201), National Natural Science Foundation of China (U21A2040), National Natural Science Foundation of China (41877520), National Natural Science Foundation of China (42077443), Industrial Technology Research and Development Project of Development and Reform Commission of Jilin Province(2021C044-5), Key Scientific and Technology Research and Development Program of Jilin Province (20200403065SF), and Construction Project of Science and Technology Innovation Center (20210502008ZP).

**Conflicts of Interest:** The authors declare no conflict of interest.

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
