# Peer review of "Three-Dimensional Vulnerability Assessment of Peanut (Arachis hypogaea) Based on Comprehensive Drought Index and Vulnerability Surface: A Case Study of Shandong Province, China"

_remotesensing, doi:10.3390/rs14215359_

Round 1
Reviewer 1 Report
General comments
This research provides important information/data worth publishing for international readers. Drought related to meteorological disasters is worthy of attention, especially for agricultural production and food security. This paper constructed a drought index suitable for peanut from the perspective of "atmosphere-plant-soil" continuum, and proposed a three-dimensional vulnerability assessment method based on this. This is very interesting and provides a new idea for crop vulnerability assessment. In addition, the questions and assumptions raised in this paper have also been clearly clarified. The method used is appropriate and reasonable, and the conclusion is credible. I like this article very much. I suggest receiving the manuscript after minor revision.
1. In section 2, the title is “Materials and Methods”. However, it only describes study area and data sources in sub-section 2.1. I think the title of section 2 should be “study area and data sources”, and section 3 is “Methodology”.
2. Page 11 Lines182-184: “Where: VCI stands for vegetation growth, which is the index value of each pixel, and, respectively, stand for the maximum and minimum NDVI of each pixel in each growth stage in recent 20 years.” Please check it.
3. In order to improve the comparability between the figures of the same drought index, it is recommended to unify the numerical range of the legend in Figure 3 (Page 10 Line350).
4. Page 11 Line393 Figure 4:Pearson correlation between MFDI and drought is weak in early growth stage, and more importantly, it is negative in some areas. Why?
5. Page 12 Line 408 Figure 5:Try to increase the resolution of Fig. 5.
6. Page 13 Line 438 Figure 6:The drought situation in the southeast of the study area in the late growth stage is quite different from that in the early and middle growth stages. Why? And what are the units of drought duration, drought intensity, and drought frequency? Days? %? Or dimensionless?
7. Page 14 Figure7 and Figure 8:By comprehensively comparing Figure 7 and Figure 8, it can be found that the drought intensity in Figure 7b is negative, but it is positive in Figure 8. Why?
8. Conclusions need to be further refined in the last section.
9. Please carefully revise all listed references. Revise all aspects, paying particular attention to abbreviations of journal titles, the ways names are listed, the addition of DOI numbers for journal papers, and web links, where possible, for other publications.
Reviewer 2 Report
The aim of this paper was to assess the vulnerability of peanut to droughts in the Chinese Province Shangdong. The authors found that the weighted drought index used in the study could better describe droughts compared to single indices; drought severity showed different patterns in the study region as a function of the growing stage of peanut; furthermore, the vulnerability of peanut to droughts was also assessed in the study.
Generally, the topic of the paper is relevant, but in my opinion, there are several issues (methods, novelty, reproducibility, formulation/English, etc.) which I think should be crucial to address.
Specific comments
-The general aim and novelty of the paper are unclear. The goals of the manuscript should be revised – they should be novel and realistic, formulated in a clear way.
-I was not able to understand how the aim of the study is linked to peanut, how the areas covered by peanut/used for peanut production were separated? To my understanding, based on Figure 1, the entire Shangdong province is investigated in the study - there is no information on land use in the study area description – is the entire province covered by peanut? Why are cities presented on Figure 4? Are the cities covered by peanut?
-If the study is based on analysing droughts for areas used as agricultural areas where peanuts are grown, why is this plant not described in detail in the study?
-Introduction: currently this section is a brief summary/description of methods in drought assessment/vulnerability assessment, etc. Instead, this section should provide a review of existing studies, it should create the context of the paper. Relevant studies should be listed. Research gaps should be identified. This section should also justify why this paper is submitted to the journal Remote Sensing. Currently I could not find in the Introduction section why this manuscript would be of interest for the readers of this journal.
-It might be a typo in Line 138, but Methods section seems to be missing?
-(possibly) Methods section: unclear description and formulations should be revised. Methods are not presented in a clear way. Please see the attached annotated pdf.
-Data section: Table 1 should be described in the text. How were all these data processed? Data used in this study should be described in detail.
-Results based on the methods section are not reproducible. Please see detailed comments in attached annotated pdf.
-The study may benefit from major English language editing; there are several unclear formulations, which are very hard to understand, it is very hard to follow the paper. Informal sentences should be avoided and revised (e.g. Line 48, 89, etc.).
-Discussion section is missing (the combined results and discussion section does not contain enough discussion, i.e. parts which put the findings of the study into the context of existing literature).
-For smaller, technical and more detailed comments please see the attached annotated pdf.

Reviewer 3 Report
The submitted paper by Wei et al. entitled: "Three-dimensional vulnerability assessment based on multi-source data fusion drought index and vulnerability surface: A case study of peanut ", deals with the vulnerability assessment of peanut crops in Shandong province, China. The manuscript looks attractive because it is well organized and presented and it deals with a challenging topic. In my opinion, the authors focused mainly on the part of the methodology. Thus, this approach creates many questions for the readers. There are some missing points that I would like to mention for the improvement of the current manuscript and its future publication.
1. The authors have described Section 3 (starting with Line 138) as results. I could not see any results in this section.
2. The authors have claimed in line 109 the construction of a “relatively perfect comprehensive” drought index. May be, authors could prove the claim by comparing the results with other studies in the region and show that this approach has performed better in predicting drought conditions in the region. It is worth noting that the manuscript should add some discussions of previous studies in the region.
3. I would suggest adding limitations to this study.
4. Table 1. Kindly add the spatial and temporal resolution of the data sets used in the study.
Round 2
Reviewer 2 Report
This is the revised version of the manuscript by Wei et al., Three-dimensional vulnerability assessment of peanut (Arachis hypogaea) based on comprehensive drought index and vulnerability surface: A case study of Shandong Province, China.
I found the authors made efforts to improve their manuscript.
However, my major comments were not addressed:
-English needs to be revised, the manuscript contains several unclear formulations, it is very hard to understand what the authors mean. There are several typos, incorrect sentences, etc.
-In the abstract, and at the end of the introduction, the authors should very clearly formulate what is new. What is novel? And what are the main, novel goals of the paper.
-Data used in the study and how they were processed should be explained in detail. What does "etc" mean in Line 143?
-Land use maps are still missing.
-Description of methods remained unclear.
-And most importantly: the study design is not appropriate. The entire Shangdong province was analysed. Results are presented on each figure for the whole province. Unless the whole province is covered by peanut - it is not right, it is not correct to present results and draw conclusions on only peanut covered areas. The text should be completely reformulated.
